# Magnetic Dilution as a Direct Method for Detecting and Evaluation of Exchange Interactions between Rare Earth Elements in Oxide Systems

**Natalia Chezhina** *[ID] and **Anna Fedorova**

Institute of Chemistry, State University, 199034 Saint Petersburg, Russia; avfiodorova@gmail.com
* Correspondence: chezhina.natalia@gmail.com; Tel.: +7-921-9212330

**Abstract:** This work is devoted to the study of exchange interactions between rare earth atoms in the $LaAlO_3$ matrix. Using the magnetic dilution method, the study of concentration and temperature dependences of magnetic susceptibility and effective magnetic moments of diluted solid solutions the magnetic characteristics of single rare earth atoms and the character of superexchange between them are described—antiferromagnetic at low concentrations, and for samarium, predominantly ferromagnetic within greater clusters as the concentration increases. The development of superexchange is similar to the exchange between d-elements in the same matrix.

**Keywords:** rare earth elements; magnetic susceptibility; magnetic dilution; superexchange

## 1. Introduction

In modern solid-state chemistry, success is observed and associated with obtaining complex oxides showing unique physical and chemical properties [1–4]; the effects of colossal magnetoresistance [5–8], nonlinear optics [9–11], transport properties [12–15], spin ordering, etc. are among them. It is well known that f-elements make a considerable impact on physical and chemical properties of materials based on complex oxides. Perhaps the most popular complex oxides used for various applied purposes are complex oxides with perovskite or perovskite-like structure.

The structure of ideal cubic perovskite ($ABO_3$) (Figure 1) defined as a cubic close packing of oxygen atoms, 1/4 of which are replaced by the so called large atoms—Ca, Sr, and Ln.

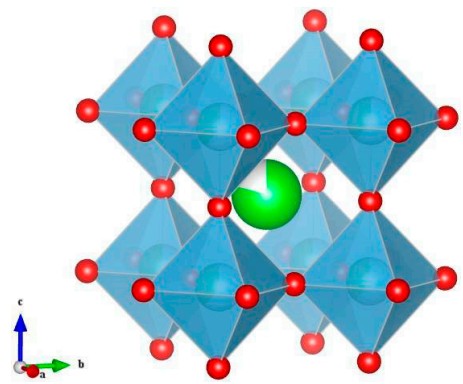

**Figure 1.** Structure of ideal cubic perovskite.

The structure of perovskite has an increased tolerance for the introduction of various doping atoms into the crystal lattice [16–19] and preserves its stability within the doping element concentration range of up to 20 mol % [20,21]. In the case of higher concentrations,

a phase stratification is often observed, which is associated with the impact of the steric factor, determined by the disproportion between the radii of the doping atoms and the sizes of the sites in the perovskite structure. This is called a structure tolerance factor [20],

$$t = \frac{r_A + r_B}{\sqrt{2}(r_B + r_O)} \tag{1}$$

where $r_A$, $r_B$, $r_O$ are the radii of *A*, *B*, and *O* ions is in the respective sites of $ABO_3$ structure. Real perovskites can have orthorhombic or rhombohedral structures when *t* deviates from 1 (0.8–1.2), especially upon doping complex oxides with elements with larger or smaller radii. This results in some changes in the angles and, to a small extent, influences the interatomic distances, however, can have an impact on the exchange channels [21].

In spite of a large scope of studies of perovskite ceramics nowadays, major problems are experienced with the lack of unambiguous concepts and theories of the mechanisms of the influence of the doping elements nature on the physical and chemical properties of doped perovskites.

An unusual behavior of oxide ceramics containing transition element atoms is accounted for by cooperative electronic effects that mostly appear in magnetically concentrated systems. Additionally, only these complex electronic effects hamper the fundamental studies of functional materials and make it impossible to track the influence of the nature and concentration of a doping M element on the properties of $A_{1-x}M_xBO_3$ perovskites [22,23].

A substitution of a fraction of various rare earth elements (Ce, Sm, Eu, Yb, etc.) for a fraction of A atoms is another method of changing the properties of ceramics, resulting in new properties with practical significance [22–27].

Therefore, their role in the electronic structure of such materials is to be revealed. Since the noted materials are used at rather high temperatures, far from zero K, the possibility of magnetic exchange interactions at about room temperature must be explored.

For a long time it was accepted that rare earth elements do not take part in magnetic exchange due to so called lanthanide compression [28]. Nowadays, however, the literature appears to explain the discrepancies in the physical properties of rare earth elements containing oxide systems at very low temperatures by some kind of exchange between rare earth elements [29].

As we deal with perovskites in this work, we must consider the possibilities of exchange interactions in such complex oxides between rare earth elements occupying the sites with coordination number 12. The work of D. Petrov and B. Angelov [30] has not been considered since the authors attempted to calculate only the direct interactions in perovskite; however, the distance between neighboring Ln atoms in this structure is about 0.53 nm. According to the data given in [31], the relative charge density of radial parts of 4f-electrons is so small that there can be no direct overlapping between orbitals. A transfer to 5d-orbitals is too expensive from the point of view of energy; however, the oxides with perovskite-like structures are well known for their superexchange interactions when d-elements appear in the B-sites of $ABO_3$ structure. It is quite possible that rare earth elements can interact between each other via oxygen atoms. The distance to the nearest oxygen atom is about 0.38 nm, and the overlapping must be essential taking into account the fact that Ln atoms do participate in the chemical bond with oxygen atoms. Of course, from the space distribution of p-orbitals of oxygen and f-orbitals of f-elements, we cannot expect a direct overlapping, but as was shown in [32,33], chromium (III) atoms that only have $t_{2g}$ occupied orbitals and took part in the $d_\pi$-$p_\pi$ interactions with a reasonable exchange parameter of 18 cm$^{-1}$ in $LaAlO_3$. Therefore, it is also reasonable to assume that such kind of interactions would be possible between f-orbitals of Ln and p-orbitals of oxygen.

A direct method of detecting and evaluating the exchange interactions in solids is magnetic dilution method—the study of magnetic susceptibility of diluted solid solutions of a paramagnetic oxide in an isomorphic diamagnetic matrix [34]. The method allows the electronic state of paramagnetic atoms and the exchange interactions in small clusters

to be detected and evaluated. Since during studying the magnetic dilution of colossal magnetoresistors ($La_{0.67}Sr(Ca)_{0.33}MnO_3$ in $LaAlO_3$ with perovskite structure [34,35] we found that the introduction of f-elements into lanthanum sites results in drastic changes in the magnetic behavior of the solid solutions (Figure 2) [36,37], we undertook the magnetic dilution of $LnAlO_3$ in the same $LaAlO_3$ matrix with the aim of revealing the magnetic behavior of rare earth elements (Ce, Eu, Gd, Yb) in the diluted solid solutions [37–40]. Our latest study is that of the magnetic susceptibility of the solid solutions of $LaAlO_3$ doped with samarium atoms—$La_{1-y}Sm_yAlO_3$ (y = 0.02, 0.20).

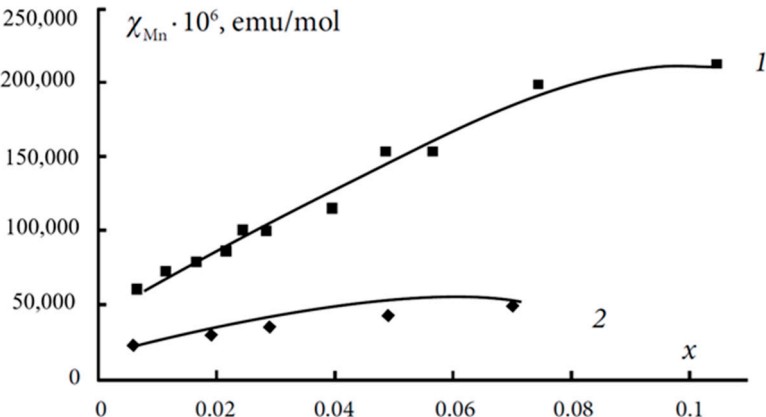

**Figure 2.** Plots of paramagnetic component of magnetic susceptibility calculated per 1 mole of manganese atoms vs. $x$ for the $x(La_{0.9}Yb_{0.1})_{0.67}Ca_{0.33}MnO_3)-(1-x)LaAlO_3$ (1) and $xLa_{0.67}Ca_{0.33}MnO_3-(1-x)LaAlO_3$ (2) solid solutions [35].

## 2. Materials and Methods

Solid solutions $La_{1-y}Sm_yAlO_3$ (y = 0.02, 0.03, 0.05, 0.07, 0.1, 0.15, 0.20) were obtained by the same procedure as the previous systems [40,41], i.e., by sol-gel method with the intermediate formation of citrate gel. The reagents in use were special pure grade lanthanum and samarium oxides, and γ-oxide of aluminum obtained under thermal decomposition of analytical pure grade $Al(NO_3)_3 \cdot 9H_2O$. Stoichiometric quantities of the starting compounds were dissolved in nitric acid (1:1), the solution was boiled down, its acidity was decreased with ammonia to a pH close to 7. Then, citric acid and ethylene glycol were added in the ratios: $n_{(CH2OH)2} = n_{C6H8O7} = \sum z_i n_i$, where $z_i$ is the i-cation charge, and $n_i$ is the number of moles of the i-cation.

The obtained gel was sintered to 800 °C with a very slow increase in temperature. After removal of organic components, the powder was pelleted in an organic glass mold. The pellets were sintered at 1450 °C for 45 h to obtain single phase samples with magnetic susceptibility not dependent on the time of additional sintering.

The content of samarium in the obtained solid solutions was determined with the help of atomic emission spectroscopy with inductively bound plasma on an ICP-AESOptima 7000 DV, Perkin Elmer (Waltham, MA, USA). The error of samarium determination did not exceed 2% from y in the solid solution formula.

The single phase of the obtained samples was proven by the X-ray analysis on a Rigaku MINIFLEX powder diffractometer using $CuK_\alpha$ emission. The powder diffractograms were identified with the help of PDF2 base. The unit cell parameters were determined with the help of Rietveld full profile analysis using Bruker TOPAS® 4.2 program package.

Magnetic susceptibility of the solid solutions was measured using the Faraday method in the temperature range 77–400 K with the help of an installation in St. Petersburg State University. The accuracy of relative measurements of the specific susceptibility was 1%. The paramagnetic components of magnetic susceptibility $\chi_{Sm}$ were calculated per 1 mole of samarium atoms. The corrections for diamagnetism were introduced with regard to

the susceptibility of the diamagnetic matrix $LaAlO_3$ measured over the same temperature range. The effective magnetic moments were calculated by the Curie formula

$$\mu_{eff} = \sqrt{\frac{3k}{N\beta^2}\chi_M \cdot T} \text{ or } \mu_{eff} = 2.84\sqrt{\chi_M \cdot T} \tag{2}$$

## 3. Results and Discussion

From the results of the X-ray analysis, all the samples containing Sm, like in the previous work [37], are single-phase and have an orthorhombic structure (Table 1). The parameter *a* slightly increases and *c* decreases, which is consistent with the ionic radii of $Sm^{3+}$ and $La^{3+}$ ($r(Sm^{3+})$ = 0.124 nm, $r(La^{3+})$ = 0.136 nm [41]).

**Table 1.** Unit cell parameters $La_{1-y}Sm_yAlO_3$.

| y | a, Å | c, Å | V, Å³ |
|---|---|---|---|
| 0.0196 | 5.358 | 13.112 | 373.425 |
| 0.0288 | 5.362 | 13.112 | 373.687 |
| 0.0492 | 5.365 | 13.109 | 374.312 |
| 0.0689 | 5.368 | 13.105 | 374.613 |
| 0.0969 | 5.364 | 13.095 | 373.776 |
| 0.1465 | 5.366 | 13.092 | 373.980 |
| 0.1842 | 5.370 | 13.091 | 374.562 |

Since after 45 h of sintering the magnetic susceptibility ceases to depend on the time of the heat treatment, we may conclude that the distribution of paramagnetic centers over the lattice is close to equilibrium.

For all the solid solutions, on the basis of experimental values of specific magnetic susceptibility ($\chi_g$), the paramagnetic components of magnetic susceptibility per mole of Sm atoms were calculated by Formula (3)

$$\chi_{Sm} = 1/y[\chi_{g(y)}M_y - (1-y)\chi_g^{solv}M^{solv}] - \Sigma_i\chi_i^{dia} \tag{3}$$

where $\chi_{g(y)}$ and $\chi_g^{solv}$ are specific magnetic susceptibilities of the solution and solvent ($LaAlO_3$), respectively, $M_y$ and $M^{solv}$ are their molecular masses, and $\Sigma_i\chi_i^{dia}$ is the sum of the diamagnetic corrections for the $LaSmO_3$ compound [42]. With the aim of comparing the results obtained for other rare earth elements studied in the same diamagnetic matrix and obtained under the same conditions, we gave the paramagnetic components of magnetic susceptibility to the isotherms (Figure 3).

The dependences of paramagnetic components of magnetic susceptibility for the $La_{1-y}Sm_yAlO_3$ solid solutions are given in Figure 4.

Besides the electron state of a paramagnetic atom, the paramagnetic component of magnetic susceptibility is a function of two main factors: the distribution of paramagnetic atoms in the matrix, which depends on the so called mutual exchange energy between various atoms occupying the same sites in the structure. In an ideal solid solutions it is zero, but in regular solutions, it can be greater, resulting in some aggregation. Additionally, it is also the function of the exchange parameter *J*. If *J* is negative, we have antiferromagnetic exchange or superexchange, as it is in perovskites. This means that the susceptibility decreases as the fraction of small clusters increases. Just this is seen in the isotherms of susceptibility for all the lanthanides studied before, but for samarium, $\chi_{Sm}$ decreases only up to y~0.07. In the concentration range 0 < 0.05, the probability of the formation of clusters greater than dimers is very low, but as y increases over 0.05, larger clusters appear with *J* > 0, which results in an increase in the susceptibility. This means that we could not only detect the superexchange interactions between a rare earth element atoms but also follow their development as the sizes of clusters in the solid solutions increase. We emphasize that a similar run of magnetic susceptibility isotherms was observed for the $LaB_xAl_{1-x}O_3$

(B—Cr, Fe) [43,44], where the distribution of paramagnetic atoms and exchange parameters were calculated using Heisenberg-Dirac-van Vleck model [45].

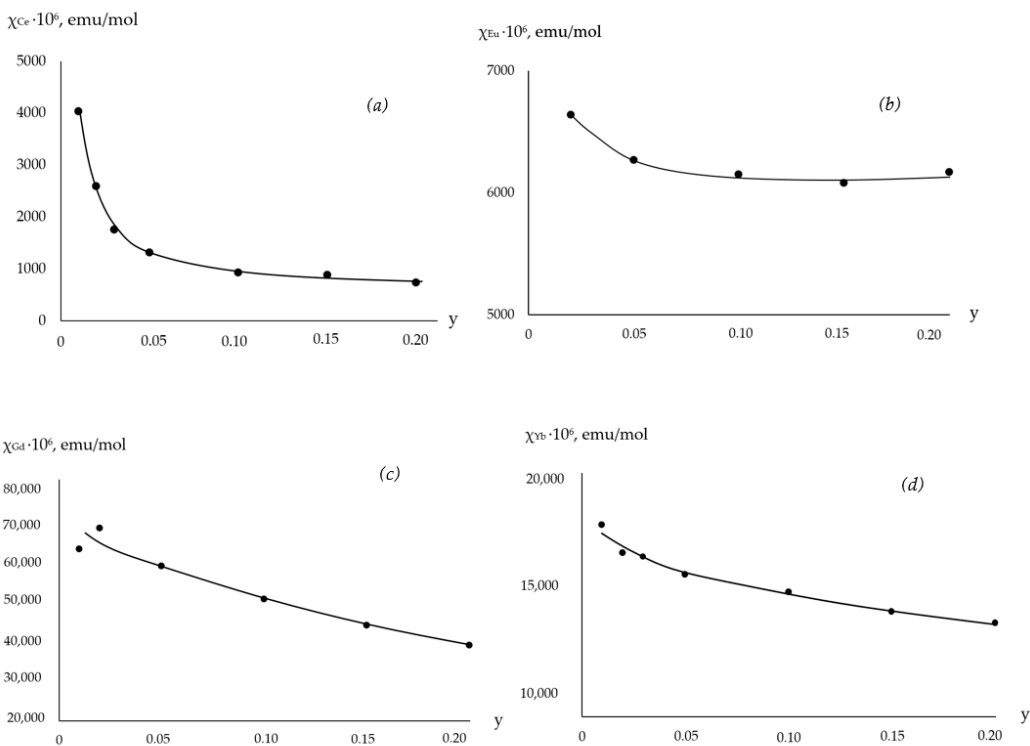

**Figure 3.** The dependences of the paramagnetic component of magnetic susceptibility on lanthanide concentrations for the $LaLnO_3$—$LaAlO_3$ solid solutions at 140 K. (**a**) Ce, (**b**) Eu, (**c**) Gd, (**d**) Yb.

$\chi_{Sm} \cdot 10^6$, emu/mol

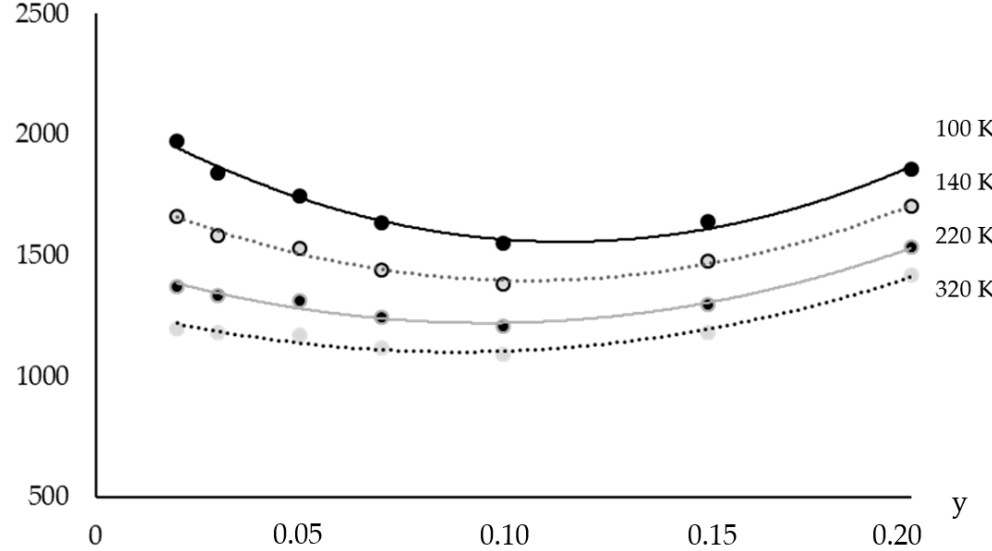

**Figure 4.** Plots of the paramagnetic component of magnetic susceptibility vs. concentration for the $LaSm_yAl_{1-y}O_3$ solid solutions.

The dependence of inverse paramagnetic susceptibility ($1/\chi_{Sm}$) on temperature is nonlinear (Figure 5).

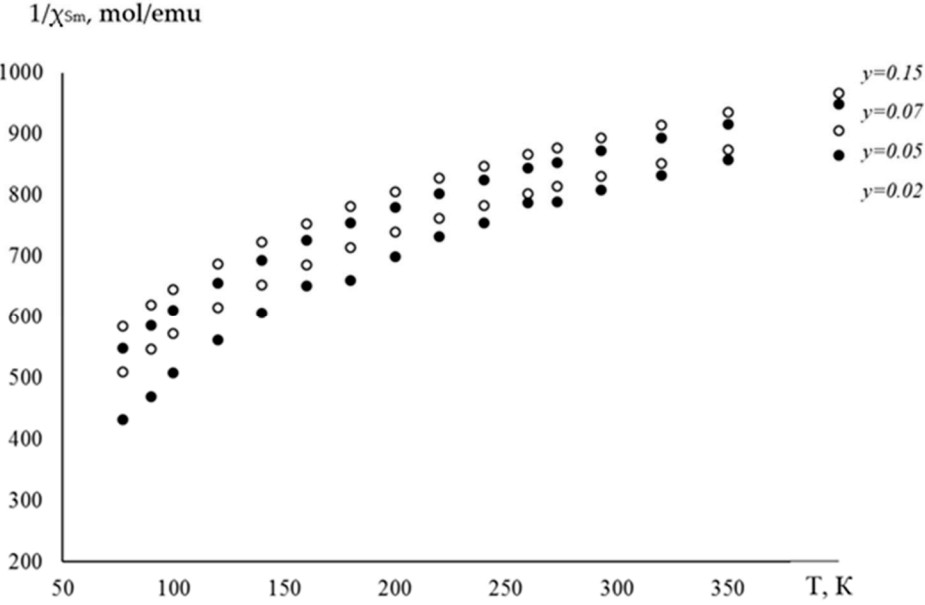

**Figure 5.** Plots of $1/\chi_{Sm}$ vs. T for the $La_{1-y}Sm_yO_3$ solid solutions.

We emphasize that the susceptibility of both samarium and europium does not obey Curie Weiss law. According to the spectral data of complexes of these metals [31] the excited states are not too far from the ground states. The authors of [31] calculated the susceptibility and effective magnetic moments with respect to occupation of all the multiplet levels—$^6H_{5/2}$, $^6H_{7/2}$, and $^6H_{9/2}$ for samarium and $^7F_0$, $^7F_1$, $^7F_2$, $^7F_3$, and $^7F_4$ for europium.

As an example we show the temperature dependences of the inverse susceptibility for ytterbium-containing solid solutions (the same for Ce), where the exited states are far from the ground state and therefore cannot be occupied (Figure 6) [31].

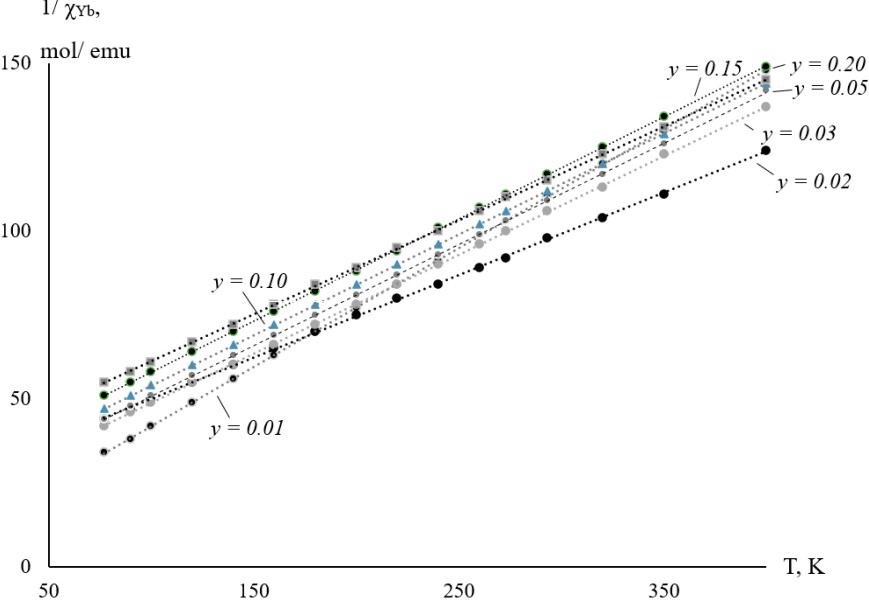

**Figure 6.** Plots of $1/\chi_{Yb}$ vs. T for the $La_{1-y}Yb_yO_3$ solid solutions.

The extrapolation of magnetic characteristics to the infinite dilution of the solid solution can give us information about the characteristics of a single lanthanide atom since it is completely impossible that any clusters of paramagnetic atoms could remain in such systems. After extrapolating $\chi_{Sm}$ and $\chi_{Eu}$ to zero concentration of the solid solutions under

study, we obtain the values of $\chi_{Ln}$ and $\mu_{eff}$, which are comparable with the results of [45]. The differences may be accounted for by various energies of the excited states in complexes and solids (Figure 7).

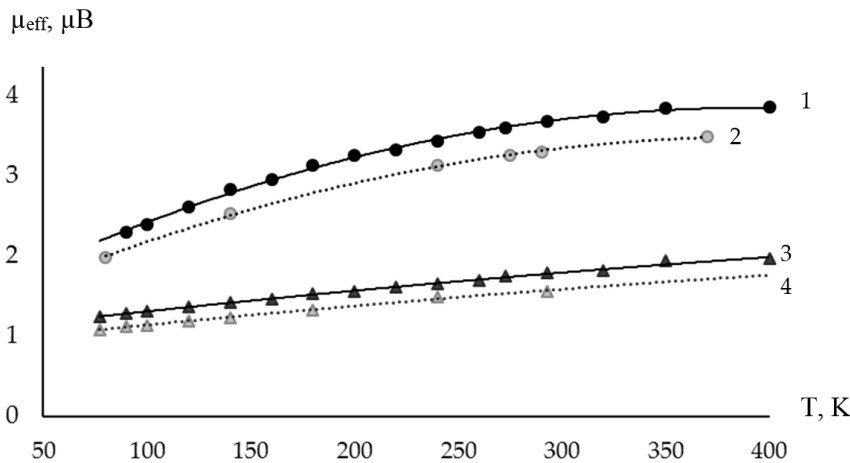

**Figure 7.** The effective magnetic moments for the solid solution containing Eu (1) and Sm (3) at the infinite dilution compared to the results of [31] for complex compounds (2—for Eu, 4—for Sm).

The theoretical effective magnetic moment for $Sm^{3+}$ calculated as

$$\mu = 2/7\sqrt{J(J+1)} \tag{4}$$

$J$ (the total moment of the ground state) is 0.84 µB. A substantial difference between theoretical and experimental $\mu_{eff}$ for the systems containing rare earth elements may be accounted for either by an admixing of excited states of the magnetic susceptibility and in a rather large temperature-independent van Vleck paramagnetism for these elements [45], or by an occupation of these nearest excited states as shown by [45]. In the first model the susceptibility is described by Formula (3),

$$\chi_{sm} = \frac{N\mu^2\beta^2}{3kT} + N_\alpha \tag{5}$$

where $N$ is Avogadro's number, $\beta$ is Bohr magneton, and $N_\alpha$ is temperature-independent paramagnetism. It is easy to determine $N_\alpha$ by plotting $\chi$ vs. 1/T. The curves are linear for both samarium and europium and give $N_\alpha$(Sm) ~0.0008 emu/mol, and $N_\alpha$(Eu) ~0.001 emu/mol. Then, after subtracting $N_\alpha$ from the susceptibility, we obtain $\mu_{eff}$ for samarium, which slightly increases with temperature and varies from 1.02 to 1.06 µB, i.e., close to the theoretical value.

In the concentration range from 2 to 5 mol. % Sm, the change in the values of the magnetic susceptibility is about 10% and cannot be explained by van Vleck paramagnetism, which is characteristic of the magnetic behavior of atoms of rare earth elements. The linear dependences of $\chi_{Sm}$ on 1/T confirm this conclusion for the $La_{1-y}Sm_yO_3$ solid solutions (Figure 8, Table 2).

We emphasize that for other f-elements studied, Ce, Gd, and Yb [39–41], the magnetic susceptibility obeys Curie Weiss low; however, the effective magnetic moment at the infinite dilution does depend on temperature, which seems to result from strong spin-orbit coupling in these elements.

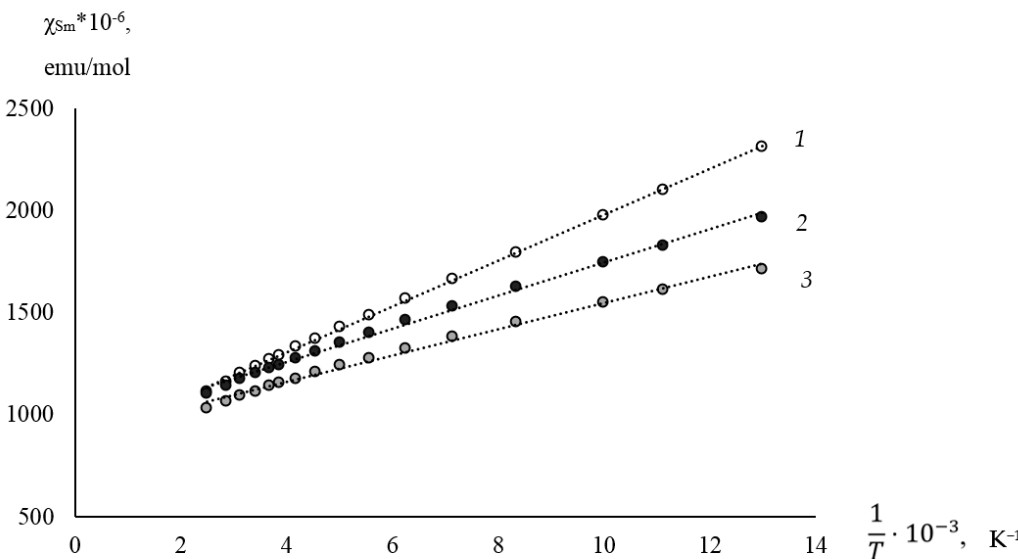

**Figure 8.** The plots of $\chi_{Sm}$ vs. $1/T$ for (1) y = 0.02, (2) y = 0.05, (3) y = 0.10.

**Table 2.** Paramagnetic component of magnetic susceptibility and effective magnetic moments for y = 0 in the $La_{1-y}Sm_yAlO_3$ solid solutions, calculated without (3) and with (5) in regard to the estimated temperature-independent paramagnetism.

| T, K | $\chi_{Sm\cdot 10}^{6}$, emu/mol | $\mu_{eff}$, µB (1) | $\mu_{eff}$, µB (2) |
|------|------|------|------|
| 90 | 2220 | 1.28 | 1.02 |
| 120 | 1900 | 1.36 | 1.04 |
| 140 | 1760 | 1.42 | 1.05 |
| 160 | 1660 | 1.44 | 1.05 |
| 200 | 1490 | 1.55 | 1.06 |
| 240 | 1380 | 1.59 | 1.06 |

## 4. Conclusions

The main conclusion following from our experimental data on magnetic dilution of $LnAlO_3$ in $LaAlO_3$ is that, in spite of the f-orbitals location close to the rare earth atom in the perovskite structure, they take part in superexchange interactions via p-orbitals of oxygen atoms. R.L. Karline [28] advocates that magnetic exchange is almost negligible for f-elements; however, if we take into account the fact that the distribution of the electron density of valent 6s- and 5d-electrons in the region of their overlapping with p-orbitals of oxygen is only larger than the electron density of 4f-orbitals by 4–5 times [37], it can be assumed that 4f-orbitals of rare earth elements take part not only in the formation of bonds with oxygen atoms but also make a substantial contribution to the magnetic superexchange. This is proved by the shape of the isotherms of paramagnetic component of magnetic susceptibility. What is important for Ce, Eu, Gd, and Yb antiferromagnetic exchange is predominant over the whole range of concentrations—0 < y < 0.2 and seems to determine the long range interactions in the concentrated systems; for samarium we see a competition between antiferromagnetic exchange within small clusters of Sm atoms, most probably dimers. In the greater clusters, the exchange becomes ferromagnetic, and the susceptibility increases. Ferromagnetic exchange may arise at the expense of mutually orthogonal f-orbitals of Ln and p-orbitals of oxygen in a cluster, which take part in the electron correlation [45]. The exchange interactions between lanthanide atoms in perovskites suggest the possibility of them taking part in the exchange between manganese atoms of the type Mn-O-Ln-O-Mn, thus resulting in drastic changes in the magnetic properties of compounds, such as doped lanthanum manganites (see Figure 2), which are promising as the basis of many important materials. Therefore, the magnetic dilution in such complex co-doped oxides must be explored further.

**Author Contributions:** The authors N.C. and A.F. contribution in the study is equal. All authors have read and agreed to the published version of the manuscript.

**Funding:** This research received no external funding.

**Institutional Review Board Statement:** Not applicable.

**Informed Consent Statement:** Informed consent was obtained from all subjects involved in the study.

**Data Availability Statement:** Fedorova, A.V.; Chezhina, N.V. Problems of Electron Structure of Colossal Magnetoresistors. In Electronic Structure of Materials. Challenges and Developments; Chezhina, N.V., Korolev, D.A., Eds.; Pan Stanford Publishing: Singapore, 2019; pp. 59–95.

**Conflicts of Interest:** The authors declare no conflict of interest.

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
