# Peer review of "Magnetic Dilution as a Direct Method for Detecting and Evaluation of Exchange Interactions between Rare Earth Elements in Oxide Systems"

_magnetochemistry, doi:10.3390/magnetochemistry9050137_

Round 1

Reviewer 1 Report

In this article, the magnetic parameters, the effective magnetic moment and exchange interaction coupling, of the rare-earth Sm doped in the perovskite oxide LaAlO3 estimated by magnetic susceptibility measurements are reported. As a notable result, the contribution of the ferromagnetic interaction in rather high Sm-concentration region of y > 0.05 is found, even though the antiferromagnetic interaction is dominant in low Sm-concentration region of y < 0.05 as in other rare-earth doped systems.

The experimental results and their analyses and interpretations seem to be properly performed, however, there are some insufficient points. Especially, the argument of the magnetic interaction based on the concentration dependence of the isothermal susceptibility is insufficient and can be misestimated. The concentration variation of the isothermal susceptibility in the Sm-doped sample is very small, order of 10-4 emu/mol, and it can be also interpreted as change of the Van-Vleck susceptibility on Sm-concentration. The split-energy between the different J-multiples of Sm3+ ion is rather small and the Van-Vleck susceptibility can be sensitive to the local distortion, for instance, and can vary sensitively on Sm-concentration. To verify the magnetic interaction between Sm-moment more properly, the temperature dependence of the susceptibility of each concentration sample is analyzed by the sum of the Curie-Weiss and Ven-Vleck terms, N*mu2*beta2/3k*(T-theta) + N*alpha, and the concentration variation of the Weiss temperature theta should be examined.

Reviewer 2 Report

Please see the attached document for comments.
